# Effectiveness of Nonpharmacological Behavioural Interventions in Managing Dental Fear and Anxiety among Children: A Systematic Review and Meta-Analysis

**DOI:** 10.3390/healthcare12050537

**Published:** 2024-02-23

**Authors:** Sarrah S. F. S. Almarzouq, Helene Chua, Cynthia K. Y. Yiu, Phoebe P. Y. Lam

**Affiliations:** 1Paediatric Dentistry, Faculty of Dentistry, The University of Hong Kong, Hong Kongckyyiu@hku.hk (C.K.Y.Y.); 2National Healthcare Group Polyclinics, Singapore 308433, Singapore

**Keywords:** dental anxiety, systematic review, nonpharmacological intervention, anxiety management

## Abstract

Background: Non-pharmacological behavioural interventions (NPBIs) have been employed by dentists to alleviate dental fear and anxiety (DFA) among preschool and school children. The aim of this systematic review and meta-analysis was to investigate the effectiveness of different NPBIs in reducing DFA among children aged below 12. Method: A comprehensive search was conducted using four electronic databases to identify randomised controlled trials that assess the effectiveness of NPBIs among preschool and school children. Two reviewers independently screened and selected the relevant studies, evaluated the risk of bias, and extracted relevant data for qualitative and quantitative syntheses. Result: A total of 66 articles were included in the study. Except during more invasive dental procedures, the use of distraction techniques was found to result in significantly lower self-rated anxiety, better cooperation, and lower pulse rate compared to the tell–show–do method. However, inconsistent results were reported regarding the efficacy of virtual reality, modelling, visual pedagogies, tell–show–do and other NPBIs in reducing DFA among children. Conclusions: The studies exhibited substantial heterogeneity due to varying age groups, methods of implementing NPBIs, dental treatments performed, and measurement scales employed in the evaluation of DFA.

## 1. Introduction

Dental fear and anxiety (DFA) among children are considered amongst the greatest daily challenges faced by general and paediatric dentists [1]. Dental fear is a normal emotional reaction to a particular threatening stimulus in a dental situation, while dental anxiety refers to a state of apprehension that something dreadful is going to happen in relation to dental treatment [2]. It is a multifactorial and widespread problem affecting all age groups [3]. The prevalence of DFA varied considerably according to studies conducted in different countries and among different age groups. For instance, DFA ranged from 4 to 20% among preschool children [4,5,6], 8 to 23% among school children aged 6–12 [5,7,8,9], 7 to 18% among adolescents [10,11,12], and 14 to 30% in the adult population [10,13,14,15].

DFA has a significant impact on an individual’s pattern of dental service utilisation. It is associated with a delay in seeking dental treatments, and children with DFA are more likely to have more dental caries and poorer oral health [16,17]. Therefore, it poses both a problem for dentists and patients [18]. Children with dental anxiety tend to only visit the dental clinic when they are experiencing acute pain, which often leads to the need for more traumatic dental procedures. These experiences intensify their fear in subsequent visits [6,11,12,13]. Furthermore, children who have had negative dental experiences perceive stronger pain than those who have had positive experiences, further exacerbating their DFA [19]. 

Advanced pharmacological interventions, such as general anaesthesia (GA) can be costly and are associated with post-operative complications [20,21,22,23]. Treating early childhood caries of a child under GA costs on average over US $2000 [24], and the mortality rate of GA for dentistry is 1 death for every 3.5 million GAs [25]. Furthermore, some studies have reported that children who were treated under GA showed caries relapse and required GA reintervention [26,27,28,29]. 

Different non-pharmacological behavioural interventions (NPBIs) have been used by general dentists and paediatric dentists to instil more positive attitudes of the child towards dental visits. Classical NPBIs include non-verbal communication, tell–show–do, positive reinforcement and distraction [30]. Other more novel NPBIs include tell–play–do, mobile dental app, audio-visual distraction, and virtual reality-based distraction [31,32,33,34,35]. Most parents and caregivers preferred NPBIs over GA due to several reasons such as lower cost, reduced waiting time and the fear of the possible risks of the pharmacological interventions [36,37,38]. To facilitate patient-centred care, alleviate DFA among children and reduce the cost burden of GA for dental treatment, it is necessary to identify more effective NPBIs that can improve DFA among children and foster more positive attitudes. Hence, this systematic review and meta-analysis were conducted to investigate the effectiveness of different NPBI among children below the age of 12.

## 2. Materials and Methods

This review was performed and reported according to the Preferred Reporting Items for Systematic Review and Meta-Analysis (PRISMA) statement [39] (Appendix A). The review protocol was registered in PROSPERO (CRD42023383595).

### 2.1. Identification of Studies (PICO) and Eligibility Criteria

The research question was formulated according to the population, intervention-control, and outcomes (PICO) model [40].

For the population, studies included healthy preschool and school children up to 12 years old. Studies that involved children older than 12 years, or children with special needs or physical disabilities were excluded from the analysis. 

For interventions, this review included all types of NPBIs that were used for any dental procedures. The interventions included but were not limited to preparatory information, non-verbal communication, voice control, tell–show–do, enhancing control, behaviour shaping and positive reinforcement, modelling, distraction, systemic desensitisation, empathy, motivational interviewing, and hypnosis.

Two types of control groups were included in this review. The first control group being no intervention for behavioural management. The second control group being any other type of NPBIs used in the study as a control group.

The primary outcome was DFA of the child, measured directly from anxiety scales, including but not limited to the Facial Image Scale (FIS) [41], Venham Picture Test (VPT) [42], the Dental Subscale of Children’s Fear Survey Schedule (CFSS-DS) [43] and the Modified Child Dental Anxiety Scale (MCDAS) [44]. Secondary outcomes were other indirect evaluations of children’s DFA, including scales of children’s behaviours, pain levels, and behavioural scales include the Frankl Scale (FS) [2], Venham Behaviour Rating Scale (VBRS) [45], and other validated scales [46,47,48]. Pain levels scales included the Wong–Baker Faces Pain Rating Scale (WBFS) [49], Face, Legs, Activity, Cry, Consolability (FLACC) Scale [50], Visual Analogue Scale (VAS) [51], or other validated scales (3), and physiological responses in response to DFA, such as heart rate/pulse rate (HR/PR) and blood pressure [52].

This review included randomised and quasi-randomised controlled clinical studies of any duration. Trials with independent treatment arms or crossover studies were both accepted. Only studies published in English were included. Non-randomised interventional studies, surveys, review articles, and case reports were excluded.

### 2.2. Search Strategy 

A systematic search was carried out in four electronic databases (Ovid Embase, Ovid Medline, PsycInfo, Web of Science) from inception to 13 October 2022. Broad keywords were used to widen the search (Appendix A). A manual search in grey literature, Google scholar, and by screening of the reference lists of relevant studies was also performed.

### 2.3. Study Selection 

Two reviewers (S.S.A and H.C.) independently selected eligible studies based on their titles and abstracts, followed by reading of full-text articles. Cohen’s kappa coefficient (k) was used to evaluate the agreements between reviewers. Any disagreement was settled by discussion or consulting the third reviewer (P.P.Y.L.).

### 2.4. Data Extraction

Data extraction of eligible studies was performed by two independent reviewers (S.S.A and H.C.). The extracted information included study characteristics (year of publication, study design, country of studies), age of children, type of non-pharmacological interventions, and type of DFA measurement tools.

### 2.5. Assessment of Risk of Bias of Included Studies

The risk of bias of each included study was assessed independently (S.S.A and H.C) using the Cochrane risk of bias tool (RoB2) [39]. This tool included five domains to address different types of bias in (I) the randomisation process, (II) deviation from the intended intervention, (III) missing outcome data, (IV) measurement of the outcome, and (V) selection of the reported result [53]. The reviewers independently evaluated each section and classified the risk categories as “low risk of bias”, “some concerns”, and “high risk of bias”. Any disagreements were resolved in consultation with the third reviewer (P.P.Y.L.).

### 2.6. Data Synthesis

The analysis was performed using STATA software version 13.1. The fixed effects model was used for meta-analysis involving fewer than five studies, while the random effects model was used for meta-analysis involving more studies [54]. 

### 2.7. Subgroup Analysis

Subgroup analyses were carried out to assess the effect of different non-pharmacological interventions with respect to different treatment procedures, subject’s age, and study design [40].

### 2.8. Assessment of Heterogeneity

I^2^ statistics and Chi square tests were conducted to assess the heterogeneity of the data synthesised [54]. The heterogeneity was determined as substantial if I^2^ is above 60% or if the *p*-value in Chi Square test was less than 0.1 [40].

### 2.9. Assessment of Publication Bias

If there were more than ten studies included in the outcome, funnel plots was used for the assessment of publication bias [55]. Otherwise, publication bias was not evaluated for the particular outcome.

### 2.10. Assessment of Certainty of Evidence

The overall certainty of evidence were assessed by two independent reviewers (S.S.A., H.C.) using the Grading of Recommendations, Assessment, Development, and Evaluations (GRADE) approach to evaluate the certainty of evidence [56]. An overall certainty of very low, low, moderate, or high was given, based on the following domains: risk of bias, imprecision, inconsistency, indirectness, and publication bias. The third reviewer was consulted (P.P.Y.L.) in cases of disagreements [56].

## 3. Results

### 3.1. Study Selection

A total of 2370 articles were retrieved through 4 databases, with 818 articles removed due to duplication. Screening based on titles and abstracts were performed on 1553 articles. 101 full-text articles were further scrutinised, and 66 controlled trials were included in this review (Figure 1). The inter-reviewer agreement was ***κ*** = 0.978.

### 3.2. Study Characteristics

The characteristics of the included studies for this systematic review are shown in Table 1. There were 3 studies published during or before the 1990s [57,58,59], 4 studies published in the 2000s [60,61,62,63], and the majority of the included studies were published in the 2010s (*n* = 30) [31,35,64,65,66,67,68,69,70,71,72,73,74,75,76,77,78,79,80,81,82,83,84,85,86,87,88,89,90,91] and 2020s (*n* = 29) [50,92,93,94,95,96,97,98,99,100,101,102,103,104,105,106,107,108,109,110,111,112,113,114,115,116,117,118,119]. Among the 66 studies included, 44 were published in Asia [31,35,50,60,61,63,64,65,71,72,73,74,75,76,77,81,84,85,86,88,89,90,91,92,93,95,96,97,98,102,103,106,108,109,110,111,112,113,114,115,116,117,118,119], 10 in Europe (*n* = 10) [62,70,79,83,87,99,100,104,105,120], 5 in South America (n = 5) [66,67,68,69,101], 5 in North America (*n* = 5) [57,58,80,82,107], and 2 in Africa (n = 2) [78,94]. The age of the participants ranged from 3 to 12 years old. The investigated NPBIs included Tell–Show–Do (*n* = 2) [75,92], distraction (*n* = 6) [50,75,81,92,109,110], video modelling (*n* = 4) [77,80,87,88], virtual reality (n = 9) [72,74,82,85,93,100,102,109,110], and visual pedagogy (*n* = 3) [69,78,95].

### 3.3. ROBS

Two independent reviewers (S.S.A, H.C) evaluated the risk of bias across the 66 included studies using the Cochrane risk of bias 2 tool [53]; the results varied. Overall, most of the included studies were assessed to be of some concern (n = 44) [19,31,35,58,59,60,61,64,65,67,68,69,70,72,74,75,77,79,80,82,83,87,88,89,91,92,95,96,97,98,100,102,105,106,107,109,110,111,113,114,115,116,117,118,119], 10 studies were evaluated to have an overall high risk, likely due to the risk of bias in randomisation and allocation concealment [44,50,57,63,73,76,78,81,85,86]. On the other hand, another 12 studies were rated as overall low risk [66,71,84,90,93,94,99,101,103,104,108,112]. Figure 2 summarises the risk of bias for each included study.

### 3.4. Comparisons between NPBIs

#### 3.4.1. Distraction vs. Tell–Show–Do (TSD)

The effect of distraction versus TSD in reducing DFA was evaluated in 6 studies while children first received their dental examination [50,75,81,92,109,110]. The distraction technique resulted in a significant reduction in child-reported DFA, as shown in the FIS (MD: −0.55; 95% CI: −0.80, −0.30; *p* < 0.001) [75,92,109], VPT (MD: −0.56; 95% CI: −0.88, −0.23; *p* = 0.001) [75,81] and CFSS-DS (MD: −0.40; 95% CI: −0.79, −0.01; *p* < 0.043) [110]. Only one study, using the Raghavendra Mahuriu Sujata Pictorial Scale (RMS-PS), reported no difference in self-rated anxiety [50]. Despite the use of subgroup analysis by measuring scales, substantial heterogeneity was still found in most of the subgroups (FIS (I^2^ = 72.4%, *p* = 0.027); VPT (I^2^ = 79.7%, *p* = 0.026)) (Appendix B Figure A1). The certainty of evidence was rated as very low due to the moderate to high risk of bias of the included studies and inconsistency. 

In addition to the reduction in dental fear and anxiety, children also exhibited more cooperative behaviours and reported less pain when evaluated with operator-rated behavioural scales (FBRS [50,81]) and self-rated pain scales (WBFS [110], FLACC [50]). However, when compared between studies, considerable heterogeneity was also found (FBRS (I^2^ = 83.4%, *p* = 0.002)) (Appendix B Figure A1). The certainty of evidence was considered very low due to the moderate to high risk of bias of the included studies, inconsistency, and imprecision.

When measuring their physiological parameters, it was reported that children exhibited significantly lower heart rates or pulse rates (HR/PR) when the distraction technique was used compared to TSD during dental prophylaxis (MD: −0.51; 95% CI: −0.90, −0.13; *p* = 0.009) [92] and dental restorative procedures (MD: −0.62; 95% CI: −0.96, −0.27; *p* < 0.001) [75,109]. Nonetheless, substantial heterogeneity was identified between studies in the subgroup analysis of dental restorative procedures (I^2^ = 96.6%, *p* < 0.001) (Figure 3).

On the other hand, the HR/PR of children were reported to be similar between the two techniques when receiving more painful procedures, including local anaesthetic administration, pulpotomy, and stainless steel crowns (MD: −0.46; 95% CI: −0.99, 0.07; *p* < 0.091) [81] (Figure 3). 

The certainty of evidence of both comparisons was considered very low due to the high risk of bias of the included studies, heterogeneity, and imprecision.

#### 3.4.2. Virtual Reality (VR) vs. Traditional Behaviour Management

When comparing VR versus traditional distraction techniques using self-rated scales, VR showed a significant reduction in child-reported DFA when measured with VPT [93] and WBFS [102,110]. No significant reduction in the child’s DFA was found when measured with FPS-R [85], FIS [100], and CFSS-DS [102]. Heterogeneity was substantial when dental anxiety was measured with WBFS (I^2^ = 85.3%, *p* = 0.009) (Appendix B Figure A2).

When comparing VR versus traditional distraction techniques using operator-rated measures, VR showed a significant reduction in the child’s DFA when measured with HR/PR (MD: −0.64, 95% CI: −0.88, −0.41; *p* < 0.001) [72,74,82,85,100,109] (Figure 4) and FIS [110] (Appendix B Figure A2).

However, no significant reduction in the child’s DFA was found when measured with FLACC [74,82,85] (Appendix B Figure A2). Heterogeneity was substantial when dental anxiety was measured with HR/PR (I^2^ = 92.5%, *p* < 0.001) (Figure 4).

#### 3.4.3. Tell–Show–Do vs. no Behavioural Intervention

Two studies reported the dental anxiety level of children when TSD was used compared to no behavioural intervention [75,92]. The self-rated anxiety when TSD was used was significantly lower when measured with VPT [75], but not with FIS [75,92] (Appendix B Figure A3).

Inconsistent findings with substantial heterogeneity between studies were also identified when comparing the HR/PR between children receiving TSD and no behavioural interventions. Children receiving TSD had significantly lower HR compared to those receiving no behavioural intervention (I^2^ = 87.0%, *p* = 0.006) (Figure 5). The certainty of evidence was considered very low due to the moderate risk of bias of the included studies, substantial heterogeneity, and imprecision.

#### 3.4.4. Video Modelling vs. Traditional Behavioural Management

Four studies included in this analysis examined the effectiveness of video modelling compared to traditional behavioural management techniques in reducing DFA among children [77,80,87,88]. The studies by Alnamankany et al. (2014) [87] and Alnamankany (2019) [88] demonstrated that watching modelling videos prior to dental treatments resulted in significantly lower self-reported anxiety and pain levels compared to control videos that were irrelevant to dentistry. These outcomes were measured using ACDAS (Abeer Children Dental Anxiety Scale) and VAS, respectively [51,121]. Hine et al. (2019) [80] also found a significant reduction in disruptive behaviours when video modelling was utilised, as assessed by a subjective operator-rated scale. However, in contrast to the aforementioned studies, Karekar et al. (2019) [77] found no significant difference in HR between children who received therapeutic storybooks (TSD), live modelling, or video modelling. Interestingly, the TSD group exhibited a lower FIS score [41].

#### 3.4.5. Visual Pedagogy vs. No Visual Pedagogy 

When comparing the use of pictorial cues to verbal reinforcement without visual cues, no significant difference was detected in children’s anxiety when measured with CFSS-DS and VPT during dental examinations (MD: −0.22, 95% CI: −0.53, 0.10; *p* = 0.185) [69,95] and dental restorative procedures (MD: −0.25, 95% CI: −0.53, 0.02; *p* = 0.067) [69,78,95]. Heterogeneity was substantial during dental examinations (I^2^ = 46.9%, *p* = 0.170) and restorative procedures (I^2^ = 81.2%, *p* = 0.005) (Figure 6). The overall certainty of evidence regarding the effectiveness of visual pedagogy was very low due to the potential risk of bias of the included studies, considerable heterogeneity, and imprecision.

## 4. Discussion

Most of the included studies consistently reported a reduction in DFA and improved behavior in children when distraction techniques were employed, as compared to TSD [50,75,81,92,109,110]. Distraction techniques are considered safe and cost-effective procedures that enhance the overall experience for patients undergoing invasive and painful medical and dental procedures [122,123,124]. These techniques involve strategies aimed at diverting the patient’s attention away from unpleasant procedures [125]. However, the studies reviewed exhibited significant inconsistencies. These inconsistencies could potentially be attributed to the wide range of distraction techniques utilised, such as toys, lavender fragrance, music, stories, and videos. It is important to note that an ideal distractor should achieve an optimal level of engagement by incorporating visual, auditory, and kinaesthetic sensory modalities. Additionally, it should elicit an active emotional response from the patient, directing their focus towards the virtual environment and minimising their awareness of the dental setting [126]. 

In this review, VR was evaluated as a distinct intervention, and the results regarding its effects on DFA in children were inconsistent. VR can be described as a computer-generated three-dimensional (3D) environment that immerses the user in a multisensory experience, temporarily transporting them away from the real world [127]. It has gained popularity in both the medical and dental fields [128]. Nine studies reported reduced DFA, pain, and HR/PR with the use of VR [72,74,82,85,93,100,102,109,110]. VR provides an immersive visual experience through occlusive headsets, effectively blocking out real-world visual and auditory stimuli. This immersive nature of VR might help alleviate anxiety, pain, and HR in children [86]. However, wearing a large VR headset over the face could also lead to a reduction in the visual field, causing a loss of control and potentially exacerbating children’s anxiety [89].

Inconsistencies were observed in the effectiveness of modelling and visual pedagogies. One possible explanation for these inconsistencies is the wide variation in the age range of participants across the different studies. The comprehension and enactment of desired behaviours taught through modelling and visual pedagogies are heavily influenced by the cognitive abilities of children, which may be less developed in the younger age groups. Additionally, the cognitive function of children, which improves with age, can also impact their behaviours in an unfamiliar dental setting [129]. Therefore, age emerges as a significant confounding factor when evaluating the effectiveness of NPBIs.

There are other NPBIs available, including protective stabilisation techniques like the hand-over-mouth exercise and Papoose board. The choice and acceptance of various behaviour management strategies are greatly influenced by various factors such as culture, parenting style, legal obligation, and the urgency of dental needs. For instance, in the United States, protective stabilisation is commonly used for uncooperative children requiring dental treatment [130]. Yet, in the United Kingdom, it is only employed by experienced clinicians under very specific circumstances [131].

Advancements in paediatric dentistry have introduced newer NPBIs such as animal-assisted therapy (AAT), which is a noninvasive intervention that involves a specially qualified animal as an integral part of the treatment process. One included randomised controlled trial found AAT to be an effective behaviour management strategy for the current generation of children [114]. However as it is a relatively new area for scientific research; more randomised controlled trials are needed to establish specific guidelines for AAT.

Another significant confounding factor is the type of treatment administered. Invasive procedures that cause more pain are more likely to result in higher levels of DFA among children [132]. Although this review conducted subgroup analyses based on the interventions employed, the limited number of studies found prevented a comprehensive evaluation of the true effects of NPBIs. Therefore, the influence of treatment type on the effectiveness of NPBIs in reducing DFA could not be fully assessed.

Self-rated scales are commonly used to assess sensations and emotions such as DFA and pain, but their reliability may be compromised when used with children. While many scales employ Likert scales to enhance children’s understanding, these measurements still necessitate a significant level of cognitive flexibility. Children must be able to shift their attention between different options, compare and differentiate choices, and retain and consolidate information before selecting the most appropriate response. Young children below the age of four are particularly susceptible to middle bias, as they tend to choose the faces at the endpoints rather than those in between [133].

Many of the included studies also employed indirect methods, such as observing children’s behaviours and measuring their pulse rate to assess DFA. Children experiencing higher levels of DFA often exhibit more uncooperative behaviours [134]. However, it is important to note that the reluctance of these children to undergo treatments may stem from other psychological and environmental factors other than DFA [135,136]. It is worth mentioning that the assessment of behaviours are mostly rated by the operator, which introduces the risk of outcome assessor bias, as operators may not blinded to the specific NPBI being used. On the other hand, physiological responses like heart rate and SpO2 provide more objective measures of evaluating DFA [52,137]. However, the equipment or measures used in these studies may not be sensitive enough to detect subtle changes and establish a clear correlation with DFA [138]. 

The certainty of evidence regarding the effectiveness of all NPBIs evaluated in this review is compromised by several factors. These include significant inconsistencies between studies, potential risk of bias, and small sample size of the included studies. 

Future research on NPBIs should prioritise certain improvements to enhance the quality of studies in this field. Firstly, it is crucial to conduct more high-quality randomised controlled trials (RCTs) with standardised protocols for implementing NPBIs. This will ensure consistency and comparability across studies, allowing for more reliable conclusions to be drawn. Furthermore, in terms of outcome assessment and dental anxiety measurement, it is recommended to utilise physiological measurements such as heart rate (HR) and pulse rate (PR). These objective measures provide a fair and unbiased assessment of outcomes, thus enhancing the validity of the findings. Incorporating these physiological measures alongside self-reported measures can provide a more comprehensive evaluation of the impact of NPBIs on dental anxiety. Lastly, future studies should aim to include larger population sizes to increase the statistical power of the trials. This will enhance the generalisability of the results and allow for more robust conclusions to be made regarding the effectiveness of NPBI interventions. By addressing these improvements, future research on NPBIs can contribute valuable insights and further enhance our understanding of its efficacy. These enhancements will ultimately lead to more evidence-based recommendations and improved dental care practices.

This systematic review and meta-analysis followed the guidelines outlined in the Cochrane Handbook for Systematic Reviews [40] and the PRISMA guidelines for reporting [39]. The study’s eligibility and risk of bias were assessed independently, and subgroup analyses were conducted based on self- and operator-rating scales, as well as intervention types, which are noteworthy strengths of this review. However, one limitation is the possibility of excluding relevant non-English articles, although the impact of this exclusion on the findings may not be significant [139].

## 5. Conclusions

The use of distraction techniques led to significantly lower self-rated anxiety, better cooperation, and lower pulse rate in comparison to the tell–show–do method, except during more invasive dental procedures. There were inconsistent results reported regarding the efficacy of virtual reality, modelling, visual pedagogies, tell–show–do and other NPBIs in reducing DFA among children. The studies exhibited substantial heterogeneity due to varying age groups, methods of implementing NPBIs, dental treatments performed, and measurement scales employed in the evaluation of DFA.

## Figures and Tables

**Figure 1 healthcare-12-00537-f001:**
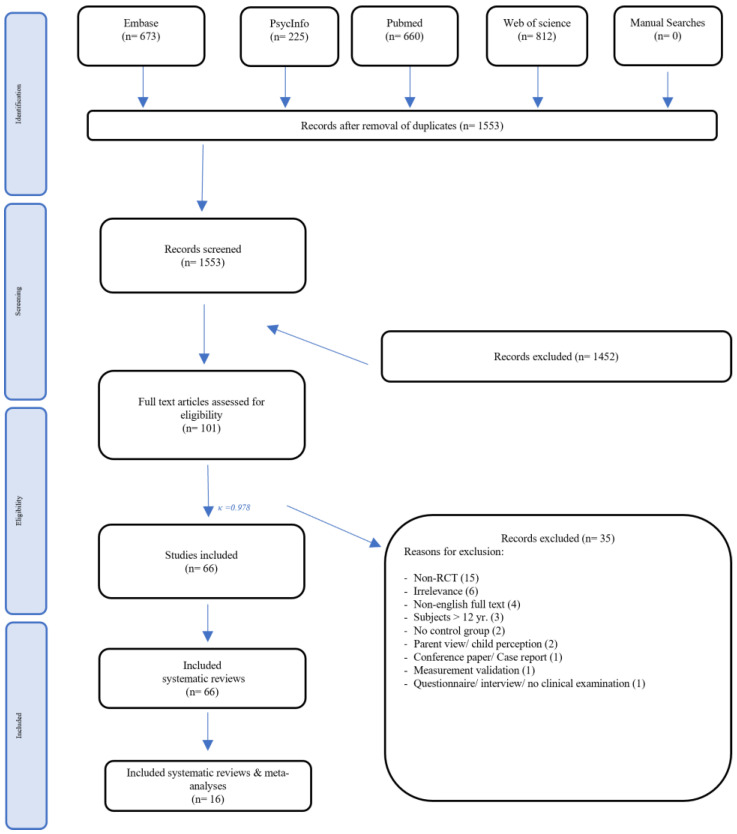
PRISMA flowchart of the current meta-evaluation.

**Figure 2 healthcare-12-00537-f002:**
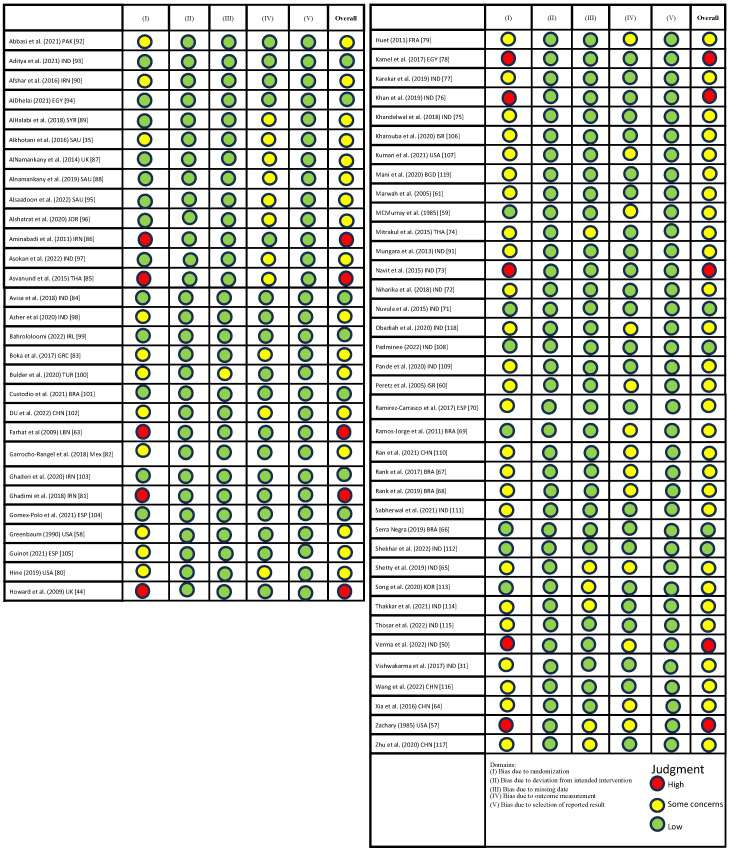
Assessment of risk of bias using ROB2: A revised Cochrane risk of bias for randomized trials.

**Figure 3 healthcare-12-00537-f003:**
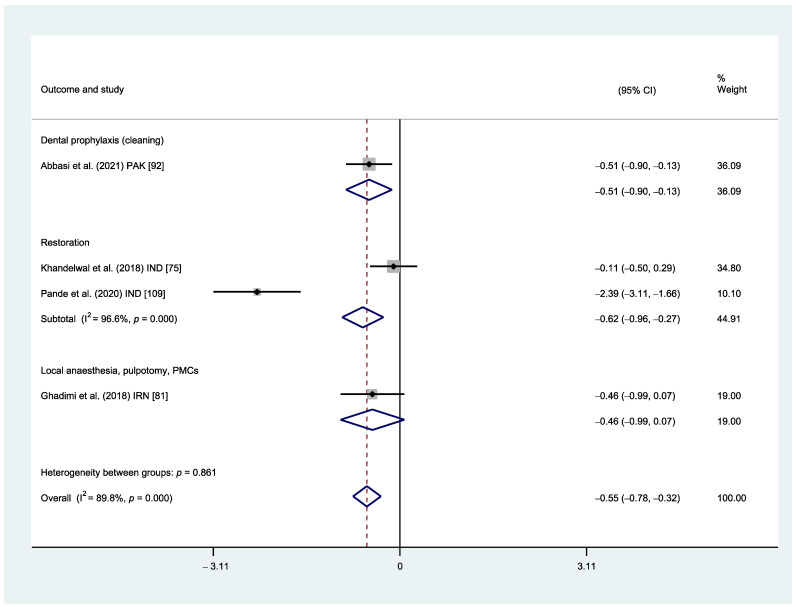
Meta-analysis subgroup analysis. HR/PR between distraction vs. Tell–Show–Do.

**Figure 4 healthcare-12-00537-f004:**
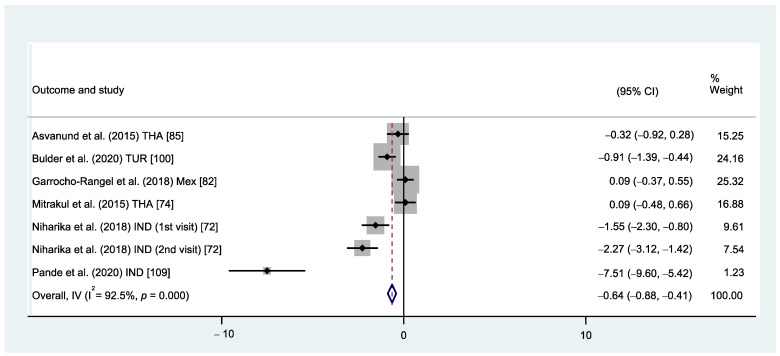
Forest plot comparison. HR/PR between VR vs. traditional behavioural methods.

**Figure 5 healthcare-12-00537-f005:**
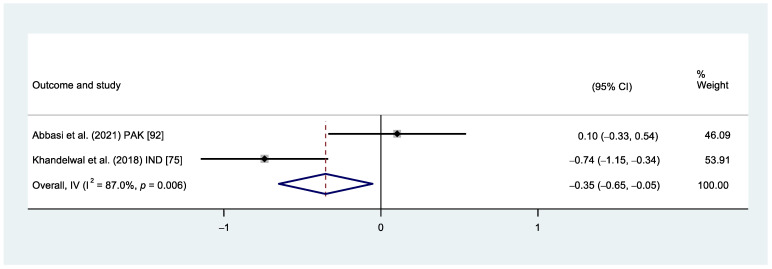
Forest plot comparison. HR/PR between TSD vs. no treatment.

**Figure 6 healthcare-12-00537-f006:**
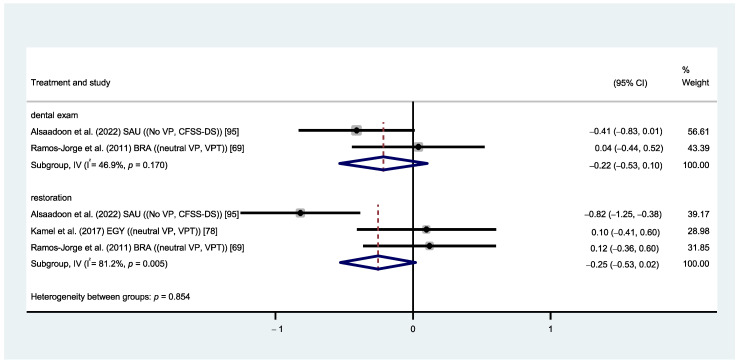
Forest plot comparison between VP vs. no VP.

**Table 1 healthcare-12-00537-t001:** Characteristics of included studies.

No.	Study (Year and Country)	Study Design, Setting	N Patient; Age Range (Years Old)	Intervention Group	Control Groups	Method of Assessment	Evaluation Time	Outcome
1	Abbasi et al. (2021) PAK [92]	RCT 4 parallel groups, dental clinic	160; 6–11	(1) Mobile application “little lovely dentist” (2) You-tube “Dental video songs” (3) Tell–show–do	No intervention	(1) Heart rate (2) Facial image scale	Pre-op/post-op	Dental anxiety
2	Aditya et al. (2021) IND [93]	RCT, 4 parallel groups, dental clinic	60; 6–9	(1) Fidget spinner (2) Kaleidoscope (3) VR Distraction	NO distraction	(1) Venham’s picture test (2) Pulse oximeter (3) Pulse rate (4) SpO2	6M	Dental anxiety
3	Afshar et al. (2016) IRN [90]	RCT, 2 (control, study) groups, dental clinic	67; 5	(1) Parental presence (2) Parental absence	No control group	(1) HR (2) Venham scale (3) Frankl BRS	Not stated	Dental anxiety
4	AlDhelai (2021) EGY [94]	RCT, 2 parallel groups, dental clinic	150; 3–6	Parental active presence	Parental passive presence	(1) FIS (2) FBRS (3) IQ level	Not stated	Child’s behaviour based on their IQ level
5	Al-Halabi et al. (2018) SYR [89]	RCT 3 groups, dental clinic	101; 6–10	Audio-visual distraction (1) Eyeglass (2) VR box or tablet	Conventional NP-BMT	(1) WBFS (2) HR (3) FLACC-BRS	6M	Dental anxiety
6	Al-khotani et al. (2016) SAU [35]	RCT 2 parallel groups, dental clinic	56; 7–9	AV distraction	NO intervention	(1) FIS (2) MVARS (3) vital signs (4) BP (3) PR	Pre-op/post-op	Dental anxiety
7	Al-namankany et al. (2014) UK [87]	RCT 2 parallel groups, dental clinic	80; 6–12	Modelling video	OH instruction video	(1) Abeer Children Dental Anxiety Scale (2) VAS	Before watching the video/after watching the video	Dental anxiety
8	Alnamankany et al. (2019) SAU [88]	RCT 2 parallel groups, hospital	46; 6–10	Modelling video	OH instruction video	Abeer Children Dental Anxiety Scale	Before watching the video/after watching the video	Dental anxiety
9	Alsaadoon et al. (2022) SAU [95]	RCT 2 parallel groups, dental clinic	93; 6–8	Received storybook	No intervention	(1) CFSS-DS (2) VCAS (3) FBRS	Pre-op/post-op	Dental anxiety
10	Alshatrat et al. (2020) JOR [96]	RCT, 2 groups, dental clinic	54; 5–12	VR distraction	No intervention	(1) VAS (2) Wong–Baker faces (3) FLACC scale	Not stated	- Dental pain - Dental anxiety
11	Aminabadi et al. (2011) IRN [86]	RCT 2 parallel groups, dental clinic	80; 6–7	Pictorial story (dentist)	Pictorial story (barbershop)	(1) Wong–Baker faces (2) MCDAS scale (3) Sound, eye and motor scale	Pre-op/post-op	- Dental pain - Dental anxiety
12	Asokan et al. (2022) IND [97]	RCT, 3 parallel groups, school	60; 4–5	1st group: magic trick distraction 2nd group: mobile dental game distraction	3rd group: TSD	Chotta Bheem–Chutki scale	Post-op	Dental anxiety
13	Asvanund et al. (2015) THA [85]	RCT, split mouth crossover, dental clinic	49; 5–8	1st visit: not wearing AV eyeglass 2nd visit: wearing AV eyeglass	1st visit: wearing AV eyeglass 2nd visit: not wearing AV eyeglass	(1) Faces Pain Scale-Revised (2) Heart rate (3) FLACC	2 visits, 1–4 weeks apart	Dental pain
14	Avisa et al. (2018) IND [84]	RCT 3 parallel groups, dental clinic	210; 8–12	(1) Acupressure (2) Sham	(1)No intervention	(1) MCDAS scale (2) Frankl (3) Pulse rate	Pre-op/post-op	Dental anxiety
15	Azher et al. (2020) IND [98]	RCT 2 parallel groups, dental clinic	48; 6–8	Bubble breath play therapy	Tell–show–do	(1) Pulse rate (2) Venham’s anxiety and behaviour rating scale	Pre-op/post-op	Dental anxiety
16	Bahrololoomi (2022) IRL [99]	RCT 2 crossover groups, dental clinic	35; 7–10	1st group: with breathing exercise 2nd group: No breathing exercise	1st group: No breathing exercise 2nd group: with breathing exercise	(1) FIS (2) BP (3) FLACC (4) WBFPS	Not stated	Dental anxiety
17	Boka et al. (2017) GRC [83]	RCT 2 groups, dental clinic	61; 3–8	Parental presence/absence + conventional NP-BMT	No PPA + conventional NP-BMT	Frankl scale	Pre-op/post-op	Childs’ behaviour
18	Bulder et al. (2020) TUR [100]	RCT 2 crossover (placebo control) groups, dental clinic	76; 7–11	1st group: - 1st visit ‘attention placebo control’ (control) - 2nd visit ‘VR’ (intervention)	2nd group: - 1st visit ‘VR’ (intervention)’ - 2nd visit ‘attention placebo control’ (control)	(1) CFSS-DS (2) FIS (3) HR	Pre-op/post-op	- Dental anxiety - Dental pain - Child’s behaviour
19	Custodio et al. (2021) BRA south [101]	RCT 2, dental clinic	44; 6–9	AV eyeglasses distraction	Conventional NP-BMT	(1) VAS (2) FLACC (3) HR (4) FPS-R scale	Pre-op/post-op	- Child’s behaviour - Dental anxiety - Dental pain - Behaviour - Body movement - Pain perception
20	DU et al. (2022) CHN [102]	RCT 2 groups, dental clinic	86; 4–9	VR relaxation	Traditional NP-BMT	(1) Modified CFSS-DS (2) Wong–Baker faces (3) Houpt scale (4) Simulator sickness questionnaire	Not stated	- Dental anxiety - Pain perception
21	Farhat-McHayleh et al. (2009) LBN [63]	RCT 3 parallel groups, dental clinic	155; 5–9	Group 1&2 ‘Live modeling’	3rd Group: Tell–show–do	HR	Not stated	- Dental anxiety - Which of the child’s 2 parents represented the model most suitable for live modeling
22	Garrocho-Rangel et al. (2018) Mex [82]	RCT 1 crossover group, dental clinic	40; 5–8	Interventional dental visit ‘Video eyeglasses/earphones system distraction’	Control dental visit ‘Tradition non-aversive behaviour management’	(1) FLACC (2) HR (3) O2 saturation	Two dental sessions	- Dental anxiety - Pain perception
23	Ghaderi et al. (2020) IRN [103]	RCT 1 crossover group, dental clinic	24; 7–9	1st group: - 1st visit ‘treated with no lavender (control)’ - 2nd visit ‘treated with lavender (intervention)’	2nd group: - 1st visit ‘treated with lavender (intervention)’ - 2nd visit ‘no lavender(control)’	Anxiety: (1) Salivary cortisol (2) PR Pain perception: (1) Face rating scale	Two dental visits	- Dental anxiety - Pain perception
24	Ghadimi et al. (2018) IRN [81]	RCT 2 crossover groups, dental clinic	28; 4–5	1st group: - 1st visit ‘cartoon distraction (intervention)’ - 2nd visit ‘tell–show–do (control)’	2nd group: - 1st visit ‘tell–show–do (control)’ - 2nd visit ‘cartoon distraction (intervention)’	(1) Venham picture test (2) PR (3) FBRS	Two dental visits	- Dental anxiety - Patient’s behaviour
25	Gomex-Polo et al. (2021) ESP [104]	RCT 2 parallel groups, dental clinic	80; 5–10	VR distraction	No distraction	(1) Facial image scale test (2) Frankl test	Not stated	- Dental anxiety - Patient’s behaviour
26	Greenbaum (1990) USA [58]	RCT, 2 groups, dental clinic	40; 3.5–4	Loud voice during tx	Normal voice during tx	(1) Dental subscale (2) Self-assessment mannequin	Not stated	Dental fear
27	Guinot (2021) ESP [105]	RCT, crossover	68; 6–8	Video game ‘PlayStation’	Cartoon film	(1) Modified Corah dental anxiety scale (2) Venham picture test (3) Wong–Baker faces scale (4) Frankl scale (5) Heart rate	10 M	Dental anxiety
28	Hine et al. (2019) USA [80]	RCT, dental clinic	40; 3–6	4 min Video modeling	14 min clip of popular children’s cartoon	(1) 15 s partial-interval recording and included physical and vocal disruptions. (2) Likert-type scale	Pre-op/post-op	Disruptive behaviour
29	Howard et al. (2009) UK [44]	RCT 2 parallel groups, dental clinic	73; 5–10	PALS model at the end of each	Motivational rewards	(1) MCDAS (2) DMFT	Not stated	- Dental anxiety - Dental caries
30	Huet et al. (2011) FRA [79]	RCT 2 parallel groups, dental clinic	30; 5–12	Hypnosis	No Hypnosis	(1) Modified Yale scale (2) VAS (3) Modified objective pain score	Over 3M	- Dental anxiety - Pain experience
31	Kamel et al. (2017) EGY [78]	RCT 2 parallel groups, dental clinic	60; 4–6	Positive images of dental treatment	Neutral cartoon images	(1) Frankl rating scale (2) Venham picture test	Not stated	- Dental behaviour - Dental anxiety
32	Karekar et al. (2019) IND [77]	RCT 3 parallel groups, dental clinic	63; 7–9	(1) Live modelling (2) Film modelling	(3) Tell–show–do	(1) FIS (2) HR	Before, during, and after diagnosis/preventive treatment	Dental anxiety
33	Khan et al. (2019) IND [76]	RCT 2 parallel groups, dental clinic	100; 4–10	AV distraction through VR Glasses 3D Box	Normal dental setup (no intervention)	(1) FIS (2) MVARS (3) BP (4) HR	Pre-op/post-op	Dental anxiety
34	Khandelwal et al. (2018) IND [75]	RCT, 4 groups, dental clinic	400; 5–8	(1) AVD (2) TSD + AVD	(1) No intervention (2) Tell–show–do	(1) FIS (2) VPT (3) BP (4) HR (5) SpO2	Before, during, and after Tx	Dental anxiety
35	Kharouba et al. (2020) ISR [106]	RCT, 2 parallel groups, dental clinic	69; 5–12	TV distraction	Tell–show–do	(1) FIS (2) Frankl scale (4) HR (5) SpO2	Pre-op/post-op	- Dental anxiety - Child’s cooperation
36	Kumari et al. (2021) USA [107]	RCT 2 parallel groups, dental clinic	100; 6–12	Immersive VR	Non-immersive VR	(1) MCDAS (2) VAS (3) WBFRS	Pre-op/post-op	- Dental anxiety - Pain perception
37	Mani et al. (2020) BGD [119]	RCT 3 parallel groups, hospital	30; 6–12	(1) Audio distraction (2) Audio-visual distraction	(3) No intervention	(1) HR (2) Venham’s picture rate (3) Venham’s clinical rating scale	1st and 2nd visits	Dental anxiety
38	Marwah et al. (2005) IND [61]	RCT 2 parallel groups, dental clinic	40; 4–8	Music distraction is divided into (subgroups) depends on the pt.’s selection: a. instrumental music group b. nursery rhymes music group	No intervention	(1) Venham’s picture rate (2) Venham’s anxiety rating scale (3) HR (4) SpO2	Four dental visits	- Dental anxiety - Type of music that is helpful in the reduction of anxiety
39	McMurray et al. (1985) [59]	RCT parallel groups, dental clinic	80; 9–12	Film model demonstrating coping strategies McMurray et al. (1985)	Film model concerned with dental hygiene	(1): Picture analogue scale (PDAS) (2) Pulse rate (3) DAI	Children were observed 1–2 week during dental examination following phycological treatment (locus of control and coping strategies)	Dental anxiety
40	Mitrakul et al. (2015) THA [74]	RCT 2 groups, dental clinic	42; 5–8	1st visit: ‘without wearing AV eyeglass’ 2nd visit: ‘wearing AV eyeglass’	1st visit: ‘wearing AV eyeglass’ 2nd visit: ‘without wearing AV eyeglass’	(1) Faces Pain Scale-Revised (2) FLACC (3) HR	- Pre-operation - RD placement - 1st use of hand-piece - 5 min interval during the remaining Tx	- Dental pain - Dental anxiety
41	Mungara et al. (2012) IND [91]	RCT 2 groups, dental clinic	90; 5–9	Film modeling	Not exposed to any film	(1) CFSS-DS	Baseline fear rating before the 1st visit and after the second visit	Dental anxiety
42	Navit et al. (2015) IND [73]	RCT 5 parallel groups, dental clinic	150; 6–12	(1) Instrumental music group (2) Musical nursery rhymes group (3) Movie songs group (4) Audio stories group	No intervention	(1) VPT (2) VCRS (3) HR	4 dental visits ‘6M’	Dental anxiety
43	Niharika et al. (2018) IND [72]	RCT 2 single blinded-crossover groups, dental clinic	40; 4–8	Group A: Session I: tell–show–do Session II: with VR Session III: no VR	Group B: Session I: tell–show–do Session II: no VR Session III: with VR	(1) Wong–Baker faces (2) MCDAS (3) HR	Three dental sessions	- Dental anxiety - Dental pain
44	Nuvvula et al. (2015) IND [71]	RCT 3 parallel groups, dental clinic, and school	90; 7–10	(1) Audio (basic technique + music) (2) AV (basic technique + 3D AV)	Basic behaviour guidance technique without distraction	(1) MCDASf (2) Pulse rate (3) Wright’s modification of FBRS and Houpt scale	Pre-op/post-op ‘7M’	Dental anxiety
45	Obadiah et al. (2020) IND [118]	RCT 2 groups, dental clinic	60; 6–12	Breathing exercise + bubble toy	No intervention	(1) Frankl behaviour rating scale (2) FIS (3) FLACC (4) Wong–Baker faces pain scale	1st and 2nd visits ‘5M’	- Dental anxiety - Pain perception
46	Padminee (2022) IND [108]	RCT 2 parallel groups, dental clinic	70; 7–12	Breathing relaxation through BrightHearts application during IANB delivery in the 1st 2 visit	VR through AV googles during IANB administration in the 1st 2 visits	(1) HR (2) Chotta Bheem–Chutki CBC scale(cartoon-based anxiety measuring scale)	3 dental visits	Dental anxiety
47	Pande et al. (2020) IND [109]	RCT 4 parallel groups, dental clinic	60; 5–8	(1) Audio distraction (2) AVD using VR (3) Mobile phone Game Distraction	(1) Tell–show–do	(1) BP (2) HR (3) FIS	Pre-op/post-op	Dental anxiety
48	Peretz et al. (2005) ISR [60]	RCT 2 groups, dental clinic	70; 3–6	Magic tricks	Tell–show–do	(1) Time from the beginning of the session to sitting on the dental chair (2) Ability to perform a dental examination (3) Frankl’s behavioural category	Pre-op/post-op	Child’s behaviour
49	Ramirez-Carrasco et al. (2017) ESP [70]	RCT 2 parallel groups, dental clinic	40; 5–9	Headphones ‘classic directive hypnosis’	Headphones to bleck out the dental drill’s noise	(1) FLACC (2) HR	During the dental visit	- Dental Anxiety - Dental pain
50	Ramos-Jorge et al. (2011) BRA [19]	RCT 2 parallel groups, dental clinic	70; 4–11	Positive image of dentistry and dental treatment	Dentally neutral image	VPT	Pre-op/post-op ‘5M’	Dental Anxiety
51	Ran et al. (2021) CHN [110]	RCT 2 groups, dental clinic	120; 4–8	VR	Tell–show–do	(1) CFSS-DS (2) WBFS (3) FBRS	- Pre-op/during - Dental procedure	- Dental anxiety - Dental pain - Compliance score in perioperative children
52	Rank et al. (2017) BRA [67]	RCT 4 groups, dental clinic	62; 4–6	(1) Mirror and conversation (2) Toys (3) Children’s stories	(1) No distraction tool	(1) FIS (2) BRS	During dental procedure For ‘6 M’	- Dental anxiety - Behaviours
53	Rank et al. (2019) BRA [68]	RCT 2 groups, dental clinic	306; 4–6	(1) a. Say–show–do b. Positive reinforcement technique with awards after dental care	(1) Say–show–do	VPT	Pre-op/post-op For ‘10 M’	Children’s motivation in two dental visits and the difference occuring between genders
54	Sabherwal et al. (2021) IND [111]	RCT 3 groups, dental clinic	60; 8–12	(1) Hypnosis (2) Progressive muscle relaxation	(1) Communication and rapport building	(1) Visual Facial Anxiety Scale (2) HR (3) SpO2 (4) BP (5) Wong–Baker faces pain scale	Pre-op/post-op For ‘5 M’	- Dental anxiety - Dental pain
55	Serra Negra (2019) BRA [66]	RCT 2 crossover groups, dental clinic	34; 4–6	1st restoration session: music 2nd restoration session: No music	1st restoration session: No music 2nd restoration session: music	(1) Pulse rate (2) EPQ-j ‘Brazilian version of the Eysenck Personality Questionnaire-Junior’	The study consisted of three consecutive clinic consultations, each lasting about 25 min, separated by intervals of 7 days.	Effect of music on children’s pulse rate
56	Shekhar et al. (2022) IND [112]	RCT 3 parallel groups, dental clinic	123; 8–12	(1) Communication with verbal positive reinforcement + stress ball ‘active distraction’ (2) Communication with verbal positive reinforcement + AV distraction ‘passive distraction’	(1) Communication with verbal positive reinforcement	(1) MCDAS (2) HR (3) Venham’s scale (4) Self-reporting and observational scale	Pre-op/post-op	- Dental anxiety - Dental pain
57	Shettty et al. (2019) IND [65]	RCT 2 parallel groups, dental clinic	120; 5–8	VR distraction	Conventional behaviour management technique	(1) MCDAS(f)r (2) Wong–Baker faces pain rating scale (3) Salivary cortisol levels	Pre-op/post-op	- Dental anxiety - Dental pain
58	Song et al. (2020) KOR [113]	RCT 2 groups, dental clinic	48; 3–7	1st treatment: ‘watched cartoon animation’ and 2nd treatment: ‘used the programme’	1st treatment and 2nd treatment: ‘pts watched cartoon animation’	(1) Heart rate (2) Wong–Baker faces Pain rating scale	Pre-op/post-op For ‘6 M’	- Dental anxiety - Dental pain
59	Thakkar et al. (2021) IND [114]	RCT 2 groups, dental clinic	102; 5–8	Pet therapy group	Conventional behaviour management technique	(1) MCDASf (2) HR	Pre-op/post-op	Dental anxiety
60	Thosar et al. (2022) IND [115]	RCT 2 groups, dental clinic	30; 4–11	1st visit: communication 2nd visit: magic thumb	1st visit: communication 2nd visit: favourite cartoon on a mobile as AV	(1) VPT (2) Modified dental analogue scale (3) HR (4) SpO2	- VPT and modified dental analogue scale were used post-op - HR and SpO2 were used pre-op, during, and post-op - For ‘3 M’	Dental anxiety
61	Verma et al. (2022) IND [50]	Pilot study, RCT 4 groups, dental clinic	80; 4–6	(1) Tell–show–do with maternal presence (2) Mobile MG (3) MG with maternal presence	(1) Tell–show–do	(1) Frankl behaviour rating scale (2) RMS-PS (3) FLACC	Pre-op/post-op For ‘6 M’	Dental anxiety
62	Vishwakarma et al. (2017) IND [31]	RCT 2 groups, dental clinic	98; 5–7	Phase I (1st visit): live modelling Phase II (2nd visit): after 7 days, subjects were subjected to rotary treatment	Phase I (1st visit): Tell–play–do Phase II (2nd visit): after 7 days, subjects were subjected to rotary treatment	(1) HR (2) FIS (3) Venham 6-point index	Pre-op/post-op	Dental anxiety
63	Wang et al. (2022) CHN [116]	RCT 2 crossover groups, dental clinic	80; 9–12	1st treatment: auricular plaster therapy (anti-anxiety) + tell–show–do 2nd treatment: auricular plaster therapy (control) + tell–show–do	1st treatment: auricular plaster therapy (control) + tell–show–do 2nd treatment: auricular plaster therapy (anti-anxiety) + tell–show–do	(1) Salivary Cortisol (2) Heart rate (3) FCS (4) MCDAS (5) Venham’s clinical anxiety obedience level rating scale	Pre-op/post-op For ‘7 M’	Dental anxiety
64	Xia et al. (2016) CHN [64]	RCT 2 parallel groups, dental clinic	100; 3–12	Reward ‘pencil eraser, a cartoon sticker, or a small notebook’.	No intervention	CFSS-DS	Pre-op/post-op	Dental anxiety
65	Zachary (1985) USA [57]	RCT 3 stratified groups, dental clinic	53; 3–11	Stress relevant film	Stress irrelevant film	(1) VPT (2) Fear thermometer (3) Palmer sweat index (4) Behaviour profile rating scale (5) Global anxiety rating scale (6) Global behaviour rating scale	Pre-op/post-op	-The effectiveness of modeling film on representative, non-clinical sample of children - the effects of stress-relevant vs. irrelevant film intervention
66	Zhu et al. (2020) CHN [117]	RCT Class-based cluster 2 groups, school	988; 7–8	Experiential learning	Tell–show–do	(1) Modified CFSS-DS (2) BP (3) HR	Pre-op/post-op	Dental anxiety

## Data Availability

The data that support the findings of this study are available from Phoebe PY Lam, S.S.F.S.A., upon reasonable request.

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
