# Peer review of "Effectiveness of Nonpharmacological Behavioural Interventions in Managing Dental Fear and Anxiety among Children: A Systematic Review and Meta-Analysis"

_healthcare, 2024, doi:10.3390/healthcare12050537_

Round 1
Reviewer 1 Report
Comments and Suggestions for Authors
The article is well described and the methodology for searching and selecting the articles in the meta-analysis is correct. The topic within the specialty of pediatric dentistry is of great value, but in some countries it may lead to discussions regarding the legal aspects of using some of the non-pharmacological methods mentioned in the text. Perhaps the authors could look into the possibility of exploring this topic in the discussion.
Comments on the Quality of English LanguageEnglish is fine, can be impruved in some points with more expecific terms of the main subject of the paper.
Reviewer 2 Report
Comments and Suggestions for Authors
Dear researchers, it has been a pleasure for me to review your article.
I would appreciate the resolution of some doubts and I would like to give you some suggestions regarding your work.
1.- Behavior management has been a highly studied topic over the years. Currently the implementation of animal-assisted activities is a hot topic. I am surprised that in their search they did not find information about it.
2.- The age range should be included as eligibility criteria and not only take into account the upper range (12 years).
3.- Please justify the years included in the review as eligibility criteria.
4.- Have only scales measuring fear or anxiety been taken into account within the eligibility criteria?
5.- I think they should take into account the calculation of sample power when selecting studies; they should reflect this as a limitation of the systematic review.
6.- Indicate the improvements to take into account for future reviews.
7.- Check the bibliographical references section, since references 2, 4, 13, 17, 21, 22, 24, 26, 29, 30, 35, 40, 41, 42, 43, 44, 45, 50, 54 , 59, 101 and 138 do not comply with the MDPI bibliographic style.
Reviewer 3 Report
Comments and Suggestions for Authors
The introduction is readable, provides a comprehensive background about the main topic, and declares the aim of the study. The authors clearly explained the rigorous methodology, which raised some questions anyway. For example, why did the authors not perform a subgroup analysis by using the type of dental treatments as a referral? Dental prophylaxis induces less anxiety than dental restoration or local anesthesia injection; therefore, the rate of success of any technique meant to reduce dental anxiety is higher in the first case than in the second. In my opinion, the small number of studies doesn’t justify such an analysis. To improve the current meta-analysis, I suggest including only the studies that performed an objective assessment of children’s dental anxiety in the quantitative analysis (e.g., heart rate, Face Rating scale, etc.). The discussion section should provide a list of suggestions to improve the quality of the studies following the current review; for example, the authors should suggest the most appropriate design, outcomes, and dental anxiety assessment based on the results of their research.
In my opinion, the study should be considered for publication after MAJOR REVISIONS.
With warm regards
Comments on the Quality of English LanguageBefore publication, the manuscript needs English editing.
Round 2
Reviewer 3 Report
Comments and Suggestions for Authors
The authors addressed my concerns about their study; in addition, the authors modified the manuscript following my suggestions. Therefore, I suggest accepting the manuscript in its current form.
With warm regards